# Surface Properties of Halloysite-Carbon Nanocomposites and Their Application for Adsorption of Paracetamol

**DOI:** 10.3390/ma13245647

**Published:** 2020-12-10

**Authors:** Beata Szczepanik, Dariusz Banaś, Aldona Kubala-Kukuś, Karol Szary, Piotr Słomkiewicz, Nina Rędzia, Laura Frydel

**Affiliations:** 1Institute of Chemistry, Jan Kochanowski University, Uniwersytecka 7, 25-406 Kielce, Poland; piotr.slomkiewicz@ujk.edu.pl (P.S.); dziewit.n@gmail.com (N.R.); laura.frydel@gmail.com (L.F.); 2Institute of Physics, Jan Kochanowski University, Uniwersytecka 7, 25-406 Kielce, Poland; d.banas@ujk.edu.pl (D.B.); a.kubala-kukus@ujk.edu.pl (A.K.-K.); k.szary@ujk.edu.pl (K.S.); 3Holycross Cancer Center, Artwińskiego 3, 25-734 Kielce, Poland

**Keywords:** halloysite-carbon nanocomposites, X-ray photoelectron spectroscopy, inverse gas chromatography, paracetamol, adsorption

## Abstract

Analysis of surface properties of halloysite-carbon nanocomposites and non-modified halloysite was carried out with surface sensitive X-ray photoelectron spectroscopy (XPS) and inverse gas chromatography (IGC). The XPS spectra were measured in a wide range of the electron binding energy (survey spectra) and in the region of C 1s photoelectron peak (narrow scans). The IGC results show the changes of halloysite surface from basic for pure halloysite to acidic for carbon-halloysite nanocomposites. Halloysite-carbon nanocomposites were used as adsorbents of paracetamol from an aqueous solution. The adsorption mechanism was found to follow the pseudo-second-order and intra-particle diffusion models. The Langmuir multi-center adsorption model described well the obtained experimental data. The presence of carbon increased significantly the adsorption ability of halloysite-carbon nanocomposites for paracetamol in comparison to the non-modified halloysite.

## 1. Introduction

Halloysite is naturally occurring a dioctahedral 1:1 clay mineral (Al_2_(OH)_4_Si_2_O_5_·nH_2_O), similar to the kaolinite, but having additionally monolayer of water molecules between the adjacent layers. The nanotubular structure of halloysite includes tetrahedral Si-O sheet forms on the outer surface of the nanotube, and a gibbsite like octahedral sheet (Al-OH) constitutes the inner surface [1]. The length of the halloysite nanotubes varies from the nanometer scale to several microns, with external diameters ranging from approximately 20 nm to 190 nm and internal diameters from approximately 10 nm to 100 nm [1,2,3]. The use of halloysite is increasing in many fields, such as clay polymer nanocomposites, catalysis, and adsorption [3]. This clay mineral exhibits the unique, one-dimensional tubular structure and such properties as high mechanical and chemical strength, and surface, which can be changed by various modifications of its surface [3]. Halloysite is highly abundant in nature, non-toxic and bio-compatible material. Recently it is used as an efficient adsorbent for water treatment, which was the subject of many adsorption studies related to the halloysite-based materials, including raw and modified halloysite and its composites with other materials [4]. Carbon materials are frequently used for wastewater treatment because of their high specific surface area, high stability, and good adsorption ability, but they are often obtained through expensive and complicated processes [5,6,7]. The nanocomposites consisting of clay minerals and carbon can be economical and effective adsorbents of water pollution in comparison to carbonaceous materials [8,9]. Clay/carbon composites are synthetized by one-pot hydrothermal process or through liquid phase impregnation and carbonization. Biomass (glucose, cellulose, or fructose) is often used as a carbon source, clay minerals, such as montmorillonite and attapulgite, are used as a matrix. These composites were used for removing phenol, 2,4,6-trichlorophenol, methylene blue, gasoline, and Cr(VI) and Pb(II) ions from an aqueous solution [10,11,12,13,14,15]. Halloysite-carbon nanocomposites exhibited significantly higher adsorption ability in comparison to common activated carbon and unmodified halloysite [16]. Jiang et al. proved that halloysite nanotubes-Fe_3_O_4_-carbon nanocomposites (HNT-Fe_3_O_4_-C) acted as a more efficient adsorbent of methylene blue than HNT/Fe_3_O_4_ and HNT [17]. Halloysite-carbon nanocomposites were obtained through liquid phase impregnation and carbonization using halloysite as the template and saccharose as the carbon precursor. They were successfully used for the adsorption of ketoprofen, naproxen, and diclofenac from an aqueous solution [18]. The presence of these pharmaceuticals in the environment is a growing problem because of their persistence and potential risk for the terrestrial and aquatic ecosystems. Surface modification of halloysite with carbon significantly improves adsorption properties of halloysite, thus thorough analysis is necessary to explain the nature of the combination of halloysite surface and the carbon layer. The surface sensitive X-ray photoelectron spectroscopy (XPS) method provides important information on sample elemental composition and is limited to sample surface (up to 10 nm), being additionally sensitive to the chemical environment of the atoms, due to high resolution of photoelectron spectra [19,20]. The inverse gas chromatography method (IGC) allows to determine the specific surface energy heterogeneity profiles and acid base properties of different materials. The knowledge about surface properties of materials, especially composites obtained through these methods, allows better prediction of their adsorption properties in relation to various adsorbates.

In the present paper, halloysite-carbon nanocomposites were studied with the XPS and the IGC methods. The studies were performed in order to investigate the influence of the surface properties of obtained nanocomposites on paracetamol adsorption from an aqueous solution by these nanocomposites in comparison to the unmodified halloysite. We have selected paracetamol as a model compounds because it is a widely used analgesic and antipyretic drug, which have been detected in wastewater, surface waters, and drinking water [21].

## 2. Materials and Reagents

Halloysite was supplied by the “Dunino” strip mine, Intermark Company, Legnica, Poland. Saccharose is a commercial product, methanol (98%) was purchased from Chempur, Piekary Slaskie, Poland. Paracetamol (acetaminophen, N-(4-hydroxyphenyl)ethanamide, C_8_H_9_NO_2_) 99% was in turn obtained from Alfa Aesar, Kandel, Germany. Methanol, acetone, acetononitrile, ethyl acetate, and dichloromethane for gas chromatography was obtained from Sigma Aldrich, Poznan, Poland. Deionized water was used in all experiments.

### 2.1. Preparation of Halloysite-Carbon Nanocomposites

Purification method of raw halloysite and its chemical composition was published in [22]. Purified halloysite (H) was used as a template and saccharose solutions (5 wt%, 10 wt%, 20 wt%, 30 wt%) were used as carbon precursor. Halloysite-carbon composites were prepared through saccharose solution impregnation of halloysite and carbonization at constant temperature 800 °C for 8 h (this temperature was obtained with a heating rate of 5 °C/min) under N_2_ atmosphere. Details are described in [18]. The obtained halloysite-carbon materials were called 5 C/H, 10 C/H, 20 C/H, and 30 C/H, respectively, for saccharose solution concentration.

### 2.2. Halloysite-Carbon Composites Characterization

The XPS measurements of the purified halloysite and halloysite-carbon nanocomposites were performed with the monoXPS system (SPECS, Berlin, Germany) [23] located at Institute of Physics of Jan Kochanowski University, Kielce, Poland. In this system, incident X-ray beam is generated by XR50M X-ray tube (Al/Ag anode) and monochromatized using FOCUS 500 quartz single crystal monochromator with 500 mm Rowland circle and rocking curve width ∼160 meV. The electrons ejected from the sample are measured using PHOIBOS 100 electron analyzer, equipped with 1D delay-line detector systems (1D-DLD). Al anode was used for the measurements, with the tube power equal 300 W (15 kV, 20 mA). Vacuum in the analyzer chamber during measurements was in the range of (6–8) × 10^−9^ mbar. The spectra were recorded with the electron analyzer pass energy equal to 30 eV. Full width in the half of maximum (FWHM) of the Ag3d_3/2_ line measured by the system was about 1.0 eV. Energy calibration uncertainty measured with conducting Ag sample was not greater than 0.2 eV. Measurements were carried out for about 1 g of halloysite-carbon nanocomposite placed in the special holder dedicated for powder samples. The powder sample of halloysite shows low electrical conductivity and during XPS measurements it is locally charged. As the result non-physical peaks related to the modification of the measured electron energy due to the sample charging can appear in the photoelectron spectra. As we have discussed in [22] the adequate charge compensation is a key factor for correct measurement of a halloysite XPS spectra. Consequently, in this work for charge compensation, a flood gun FG-500 with electron emission current I = 70 μA and electron energy E = 3 eV was used. In these conditions, the sample current was about 2.5 μA. The use of charge compensation allowed for measurement of the XPS spectra of the halloysite samples with high accuracy (~0.2 eV), which was confirmed by the subsequent analysis of the C-C peak. Consequently, all spectra shown in this article were plotted without normalization to the C-C peak.

Qualitative and quantitative analyses of the spectra were performed using the CasaXPS software (ver. 2.3) delivered with the monoXPS system The photoelectron spectra were fitted using Shirley background and Lorentzian asymmetric lineshape with tail damping (LF).

Specific surface energy heterogeneity profiles and acid base properties of halloysite and halloysite-carbon nanocomposites were determined with a surface energy analyzer (IGC-SEA) of Surface Measurement Systems Ltd. (London, UK) at low surface coverage. Experiments were performed at 423 K with a helium carrier gas flow rate of 20 cm^3^/min using a flame-ionization detector, methane gas as marker for the hold-up time and n-alkanes, methanol, acetone, acetononitrile, ethyl acetate, and dichloromethane as probe compounds.

### 2.3. Adsorption Measurements

Batch adsorption experiments were conducted in Erlenmayer’s flasks in which appropriate adsorbent and paracetamol solution were placed. The measurements were carried out at 25 °C and mixing rate 150 rpm. Next, the cup-type centrifuge was used to separate the adsorbent from the solution. Concentration of paracetamol solution (before and after the adsorption) was determined using a UV Shimadzu UV-1800 spectrophotometer, Thermo Fisher Scientific, Waltham, MA, USA. The wavelength used to determine the paracetamol concentrations was equal 243 nm.

The Equations (1) and (2) were used to calculate the removal efficiency (R, %) and the amount of paracetamol adsorbed at equilibrium (adsorption capacity, *q_e_*, mg/g):(1)$$R\;\left(\%\right)\;=\;\frac{C_{0}-C_{e}}{C_{0}}\cdot 100$$
(2)$$q_{e}\;=\;\frac{(C_{0}-C_{e})V}{m}$$
where *C*_0_ and *C_e_* (mg/L) are the initial and equilibrium concentrations of adsorbate solutions, *V* (L) is the volume of the adsorbate solution, *m* (g)—mass of adsorbent.

Adsorption equilibrium constants were determined with the inverse liquid chromatography (ILC) method using Thermo Scientific Dionex UltiMate 3000 Series chromatography system (Thermo Fisher Scientific Inc., Waltham, MA, USA) with a diode array UV detector (DAD 190–800 nm), Chromeleon software (ver. 1.0, 2013) and the KSPD software (the details are described in [24]). Adsorption measurements were performed at a temperature range of 298 K–313 K. The adsorbate concentration was identified at a wavelength UV-DAD detector of 243 nm.

Calculations of equilibrium concentration of adsorbate (*C_e_*) in the liquid phase as well as the amount of the adsorbed adsorbate (*a*) were conducted on the basis of the data concerning peak profile division contained in CDPS database [25]. As a result, relation *a* = *f* (*C_e_*) is a function that describes an adsorption isotherm. The details of conducted calculations are described in the [24].

## 3. Results

### 3.1. Characterization of Halloysite-Carbon Nanocomposites

Analysis of Total Carbon confirmed the content of carbon in the halloysite-carbon composites: 5 C/H—2.2 wt%, 10 C/H—2.9 wt%, 20 C/H—4.6 wt%, and 30 C/H—6.7 wt% [18]. Covering halloysite surface with amorphous carbon was confirmed by diffraction patterns of H, 5 C/H, and 30 C/H samples, ATR FT-IR spectra of halloysite and halloysite-carbon composites (5 C/H, 10 C/H, 20 C/H, and 30 C/H) showed the interaction between carbon as well as inner and outer surfaces of halloysite particles [18].

The XPS measurements were carried out for purified halloysite (H) and halloysite-carbon nanocomposites (5 C/H, 10 C/H, 20 C/H and 30 C/H). Example of the survey spectrum measured for the H sample is presented in the Figure 1. In the spectrum, peaks of O 1s, Si (2s, 2p), and Al (2s, 2p) were identified, which might have been expected based on chemical structure Al_2_Si_2_O_5_(OH)_4_ of a halloysite. Additionally, Fe impurities at the amount close to the detection limit of the XPS method were measured.

The presence of Fe in the survey spectra (Figure 1) is probably due to residual iron oxide (hematite) impurities in the purified halloysite [21] or the substitutions of Al in the octahedral sheet of halloysite by Fe [26]. Moreover, as for the most samples that are exposed to the atmosphere, small amount of an adventitious carbon was detected as well.

Quantitative results for the H sample and the halloysite-carbon nanocomposites (5 C/H, 10 C/H, 20 C/H, and 30 C/H) are summarized in the Table 1 (at% means atomic percentage).

Carbon content for the H sample resulted from the presence of adventitious carbon in the all studied samples. Conversion of the carbon content expressed as an atomic percentage into weight percentage and subtracting carbon content for the H sample from the carbon content in carbon-halloysite nanocomposites provided carbon amounts for carbon-halloysite nanocomposites which were as follows: 1.5 wt% 5 C/H, 3.1 wt% 10 C/H, 5.4 wt% 20 C/H, and 8.5 wt% 30 C/H. These values are comparable with the values obtained on the basis of analysis of Total Carbon, the differences may result from different methods and different errors of determinations.

The most visible change in Table 1 is related to the increasing carbon content in carbon-halloysite nanocomposites. This content changes linearly, rising according to the increasing concentration of precursor solution (5 wt%, 10 wt%, 20 wt%, 30 wt%).

In order to understand the changes in the structure of the halloysite modified with carbon, detailed measurement and analysis of the carbon peak was performed. The results of the analysis are presented in the Figure 2 for the H sample, as well as for the 5 C/H, 10 C/H, 20 C/H, and 30 C/H halloysite-carbon nanocomposites.

In the carbon peak spectra, the following components were identified: O-C=O, C=O, C-O-C (C-O-H), C-C, and ~282.6 eV component. The ~282.6 eV component corresponding to the C–Si species was identified in C1s spectrum registered for halloysite nanotubes after silane grafting [27]. However, this finding requires further investigation of possible impact on the results of inhomogeneous electrostatic charging of the composite, consisting of both dielectric clay and conductive carbon. The intensity above 290 eV is likely due to a satellite from graphite. FWHMs, binding energies and areas obtained for the identified components are presented in the Table 2. Based on the table analysis, it can be concluded that the position of the peaks (binding energies) and their FWHMs did not change significantly (are within expected uncertainty of the measurement without normalization to carbon C-C peak) for different carbon contents. One has to remember that the narrow carbon scans were not normalized to the adventitious carbon C-C peak before fitting due to the expected difficulty in determining the location of this component before making the final fit. This problem is particularly visible in the 5C/H and 10C/H spectra (see Figure 2), where the position of the carbon C-C peak can be determined only after performing a full carbon peak components analysis. However, despite the lack of the normalization, the final positions of the C-C peak turned out to be within ~0.2 eV consistent with the expected (284.8 eV) position of this line (see Table 2). Additionally, the positions of the other essential peaks identified agree well with the theoretical values C=O (288 eV), C-O-C (C-O-H) (286.5 eV) [28].

However, the Table 2 shows clearly visible changes in the intensity (peak area) of individual components, especially for the strongest C-O-C (C-O-H), and C-C, and ~282.6 eV components. These changes of the intensities as a function of the saccharose solution concentration (5 wt%, 10 wt%, 20 wt%, and 30 wt%) are shown in the Figure 3. As can be seen from this figure, the area of the C-C and Si-C components increased gradually with the saccharose solution concentration, while the peak area of the C-O-C (C-O-H) component first increased and then decreased almost to the initial value. Therefore, it can be concluded that the concentration of the saccharose solution using as the precursor of carbon has a significant impact on the formation of carbon layer on halloysite surface. Reducing the number of functional groups on carbon surface could be observed along with the increasing concentration of precursor solution.

Inverse gas chromatography (IGC) method was applied to determine the surface properties of the halloysite-carbon nanocomposites. The IGC is a useful method of evaluating adsorption isotherms, dispersion and specific interfacial energy profiles of adsorbent at low partial pressures of adsorbates.

In this study, the Schultz [29,30] approach was used to calculate the dispersive component of free adsorption energy $\gamma_{s}{}^{d}$ of halloysite with n-alkane probes.

The value of $\gamma_{s}{}^{d}$ can be determined from Equation (3):(3)$$RT\cdot ln\left(V_{N}\right)=2N_{A}\cdot a_{CH_{2}}\cdot \sqrt{\gamma_{s}{}^{d}}\cdot \sqrt{\gamma_{l}{}^{d}}+C$$
where$\;\gamma_{s}^{d}$—dispersive component of surface energy [mJm^−2^], $\gamma_{l}^{d}$—dispersive energy of probe molecule [mJ/m^−2^], N_A—_Avogadro number, cross-section area $a_{CH_{2}}$ of the alkane molecule is calculated using the following equation: aCH2=1.09×1014·Mρ·NA23, C—constant value.

The dependence $RT\cdot ln\left(V_{N}\right)=f\left(a_{CH_{2}}\cdot \sqrt{\gamma_{l}{}^{d}}\right)$ is a straight line. The value of dispersive free surface energy for the solid phase can be determined from the value of slope. Polar probes loaded into the column with the adsorbent in identical conditions as for n-alkanes will not be in the n-alkane line. A component of specific adsorption energy $∆G_{a}^{sp}\;$can be received from the vertical distance on the axis $RT\cdot ln\left(\;V_{N}\right)$ from the n-alkane line to the point of polar substance.

IGC method also allows assessing acidic-basic properties of adsorbents. For this purpose polar substances with known donor-acceptor properties must be used. The polar probes may be acidic (electron acceptors), basic (electron donors), or amphoteric.

A Gutmann equation (Equation (4)) [31] is used for the calculations of acidic and basic characteristic of solids:(4)$$-∆G_{a}^{sp}=K_{a}\cdot DN+K_{b}\cdot AN^{*}$$
where $K_{a}$ is the acidic characteristic of solid, $K_{b}$ is the basic characteristic of solid, DN is the donor number, $AN^{*}$ is the modified acceptor number, expressed by the equation: $AN^{*}=0.288\left(AN-AN^{d}\right),\;$where $AN^{d}$ is the part of the forces of dispersion in the value of the acceptors number, which may be calculated on the basis of the surface tension measurement; constant 0.288 refers to warm interaction between the acid–base SbCl_5_ and Et_3_PO and displacement spectra for this system with respect to the transfer Et_3_PO in hexane, and AN is the acceptor number. $AN^{*}\;$is expressed in the same scale and with the same units as DN. The calculated values of $AN^{*}$ take into account the amphoteric character of the molecules [32].

From the dependence $\frac{∆G_{a}^{sp}}{AN^{*}}=f\left(\frac{DN}{AN^{*}}\right)$, constant $K_{a}$ can be calculated from the directional coefficient of the plot; the value $K_{b}$ can be calculated using OY axis intersection. Ratio $\frac{K_{b}}{\;K_{a}}$ enables specifying the character of the adsorbent surface. If the ratio is $\frac{K_{b}}{\;K_{a}}>1$, then the surface is basic (donor properties prevail over the acceptor ones). If, however, the ratio is $\frac{K_{b}}{\;K_{a}}<1$, then the surface is acidic. On the other hand, if $\frac{K_{b}}{\;K_{a}}\approx 1$, then the surface is amphotheric [33].

Based on the results of measurements with the surface energy analyzer (IGC-SEA) the calculations of the dispersive component of free adsorption energy $\gamma_{s}{}^{d}$ and specific adsorption energy $∆G_{a}^{sp}$ with the Schultz method were performed with Cirrus Plus software [34]. The Cirrus Plus software uses for calculation of the physicochemical data of probe compounds presented in Table 3.

The values of dispersive component of surface energy are 55.8 mJ/m^2^ for adsorbent H and increases in the following order: 5C/H 57.3; 10C/H 62.0; 20C/H 64.9; 30C/H 89.8 mJ/m^2^ for halloysite-carbon nanocomposites, respectively. These values are comparable to the values of$\;\gamma_{s}^{d}$ reported in the literature for carbon materials [35], which confirms that carbon is the main adsorbent component in the halloysite-carbon composite.

Surface properties of halloysite and halloysite-carbon nanocomposite samples are presented in Table 4. The obtained values of the ratio of $\frac{K_{b}}{K_{a}}$ for all nanocomposites pointing the acidic character of adsorbent surface. In the case of the non-modified halloysite (H), the value of the ratio of $\frac{K_{b}}{K_{a}}\;$confirming the basic character of adsorbent surface. Presence of carbon on the halloysite surface changed its character, the surface became acidic (acceptor properties prevail over the donor ones).

### 3.2. Adsorption of Paracetamol

Removal efficiency of paracetamol from aqueous solution on the following adsorbents: H, 5 C/H, 10 C/H, 20 C/H, and 30 C/H are showed in Figure 4. On the basis of these results it can be concluded that the halloysite-carbon composites adsorb paracetamol significantly more than halloysite (H). Removal efficiency increased with the increase of carbon content in nanocomposites reaching the highest value of 93% for the adsorbent 30 C/H.

#### 3.2.1. Kinetic Models

Pseudo-first-order kinetic model [36], pseudo-second-order kinetic model [37] and intra-particle diffusion model [38] were applied for the adsorption of paracetamol on H and 30 C/H adsorbents. Adsorption kinetics for paracetamol on H and 30 C/H adsorbents is given in Figure 5. The pseudo-first order and pseudo-second order rate constants, k_1_ and k_2_, and correlation coefficients (R^2^) are collected in Table 5. The values of correlation coefficients obtained for the pseudo-first-order and pseudo-second-order models confirmed that adsorption of paracetamol on H and 30 C/H adsorbents corresponded to the pseudo-second-order kinetic model, suggesting the chemisorption as the adsorption process.

The Weber–Morris diffusion model was applied to explain the adsorption mechanism of the paracetamol on the H and 30C/H adsorbents. The values of k_d1_, k_d2_ and c_1_, c_2_ determined from the slopes and intercepts of the first and second linear part of graph (Figure 6) are collected in Table 6. The dependency q_t_ vs. t_1/2_ is the multi-linear plot. The first part on the graph corresponds to the faster step during which the diffusion of adsorbate molecules to adsorbent outer surface occurs. The presence of the second part confirms the slower adsorption, where intra-particle diffusion is a controlling step of the whole adsorption process [38].

#### 3.2.2. Adsorption Isotherms

The fit of the experimental data was carried out using non-linear regression (Levenberg–Marquardt least square method with OriginLab software, ver. 2018) for the following adsorption models: Langmuir (one-center and multi-center) [39] and Freundlich [40] (Figure 7).

Langmuir isotherm models are represented by Equations (5) and (6) and Freundlich by Equation (7):(5)$$\mathrm{Langmuir}\;(\mathrm{one}-\mathrm{center})\;\;q_{e}\;=\;\frac{q_{m}K_{L}C_{e}}{{1+K}_{L}C_{e}}$$
where *C_e_* is the equilibrium concentration of a solute in an aqueous solution (mg dm^−3^); *q_e_* is the amount of a solute adsorbed per gram of the adsorbent at equilibrium (mg/g), *q_m_* is the maximum monolayer coverage capacity (mg g^−1^); *K_L_* is the Langmuir isotherm constant (dm^3^ g^−1^):(6)$$\mathrm{Langmuir}\;(\mathrm{multi}-\mathrm{center})\;\;q_{e}\;=\;\frac{q_{m}{\left(K_{ML}C_{e}\right)}^{1/n}}{1+{\left(K_{ML}C_{e}\right)}^{1/n}}$$
where *n* is the adsorption model index, and *K_ML_* is the Langmuir isotherm constant (dm^3^∙mg^−1^)^1/n^:(7)$$\mathrm{Freundlich}\;\;q_{e}{\;=\;K}_{F}C_{e}^{1/n}$$
where *K_F_* is the Freundlich constant (mg g^−1^(dm^−3^ mg^−1^)^1/n^) and 1/*n* is the heterogeneity factor.

The Freundlich and Langmuir (one-center and multi-center) equations parameters as well as correlation coefficients R^2^ for the adsorption of paracetamol on H and 30 C/H adsorbents at three temperatures are collected in Table 7. The highest values of correlation coefficients for adsorption on H and 30 C/H adsorbents for the multi-center Langmuir adsorption model confirmed that adsorption occurs on multiple active centers on adsorbent surface without dissociation of paracetamol molecules. The lower adsorption constant values for adsorbent H than for adsorbent 30 C/H and the decrease of these values for both adsorbents with increasing temperature indicate the exothermic character of the adsorption process. The *n* values were fractional for both adsorbents proving that the adsorption of paracetamol takes place on a different number of adsorptive centers on surfaces of H and 30 C/H adsorbents.

## 4. Conclusions

In this paper, the results of surface properties investigations of carbon-halloysite nanocomposites have been presented. Performed XPS investigations allowed us to conclude that the carbon content in nanocomposites samples increased linearly with saccharose concentration (carbon precursor). The analysis of XPS spectra confirmed the presence of the following functional groups: O-C=O, C=O, C-O-C on the surface of carbon-halloysite nanocomposites. Analysis of carbon content in carbon-halloysite nanocomposites using total carbon and XPS methods shows that the results obtained by these methods are comparable.

IGC analysis of the nanocomposites and halloysite was carried out to show that the presence of carbon on the halloysite nanotubes changes the nature of this surface, significantly facilitating the adsorption of paracetamol on the surface of the composites. Based on the IGC results, it was shown that the surface of these nanocomposites has acidic character with stronger acceptor properties, while the character of halloysite surface is basic.

The study of adsorption of paracetamol from an aqueous solution on halloysite and carbon-halloysite nanocomposite has shown that removal efficiency of paracetamol for carbon-halloysite nanocomposites was significantly higher than for non-modified halloysite. The carbon content in nanocomposites determined by XPS method increased in the same order as the removal efficiency of paracetamol. This proved that the carbon layer on the halloysite surface in the nanocomposites is mainly responsible for the adsorption of this compound from water.

Adsorption kinetics on the halloysite and carbon-halloysite nanocomposites was described with the pseudo-second-order kinetic model, suggesting the chemisorption mechanism during the adsorption process. Adsorption process of paracetamol on halloysite and carbon-halloysite adsorbents occurred according to multi-center Langmuir adsorption model (Langmuir adsorption model on multiple active centers without dissociation). Fractional values of factor *n* indicate that the paracetamol adsorption mechanisms take place with a different number of adsorptive centers on halloysite and halloysite/carbon nanocomposites surface.

Paracetamol is a weak electrolyte, which can coexist in both ionized and non-ionized form in aqueous solution. The concentration of these forms depends on the solution pH and pK_a_ (the Henderson-Hasselbalch equation). pK_a_ of paracetamol is equal 9.38. That is why about 90% of paracetamol molecules is in its protonated form up to pH 7, while in basic medium (pH 11) about 90% of the deprotonated form is found [41]. Non-electrostatic interactions with dispersion and hydrophobic character between non-dissociated paracetamol molecules and the adsorbent surface containing mainly oxygen functional groups are possible in an aqueous solution (pH 6–7). Adsorption of paracetamol on activated carbon can occur according to the π-π dispersion interaction, the H-bonds formation with surface oxygen groups, such as carboxyl or carbonyl groups or by the formation of donor-acceptor electron complexes [42]. The presence of the oxygen functional groups on the surface of carbon-halloysite nanocomposites confirmed by the XPS analysis allows us the conclusion that the adsorption of paracetamol on these adsorbents can occur according to the mechanisms described above.

The obtained surface characteristics of carbon-halloysite nanocomposites and the results of adsorption studies show that these materials can be successfully used to remove paracetamol from the aquatic environment.

## Figures and Tables

**Figure 1 materials-13-05647-f001:**
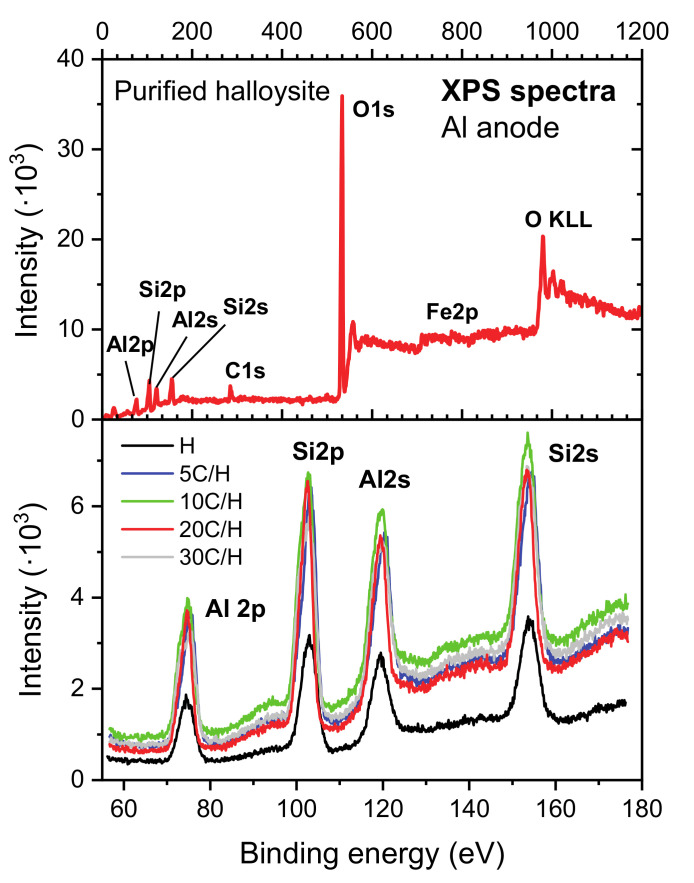
Survey XPS spectrum (top panel) of the purified halloysite sample (H). In the spectrum, O, C, Si, Al, and Fe lines were identified. In the bottom panel narrow scans of the Al2p, Si2p, Al2s, Si2s, peaks are presented. The spectra were not normalized to adventitious carbon C-C peak position (check discussion in the text).

**Figure 2 materials-13-05647-f002:**
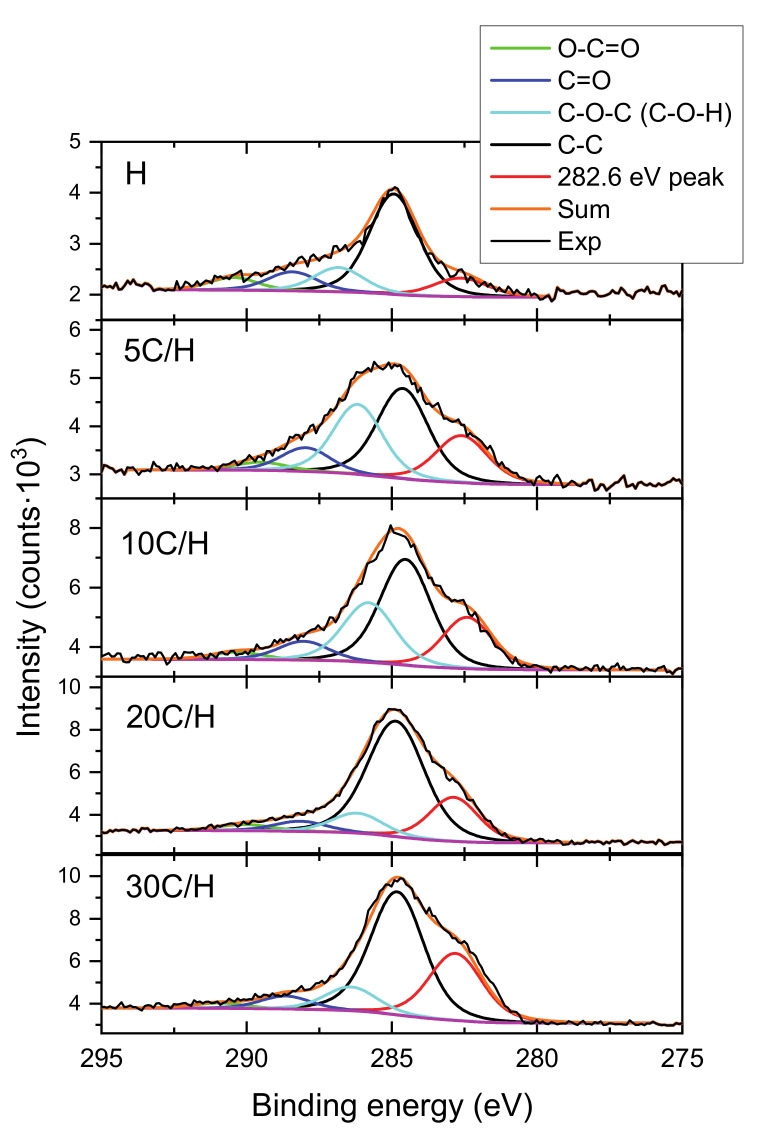
The results of the C1s peak decomposition into components corresponding to different carbon bonds, obtained for purified halloysite sample (H), and 5 C/H, 10 C/H, 20 C/H, and 30 C/H halloysite-carbon nanocomposites. The spectra were not normalized to adventitious carbon C-C peak position.

**Figure 3 materials-13-05647-f003:**
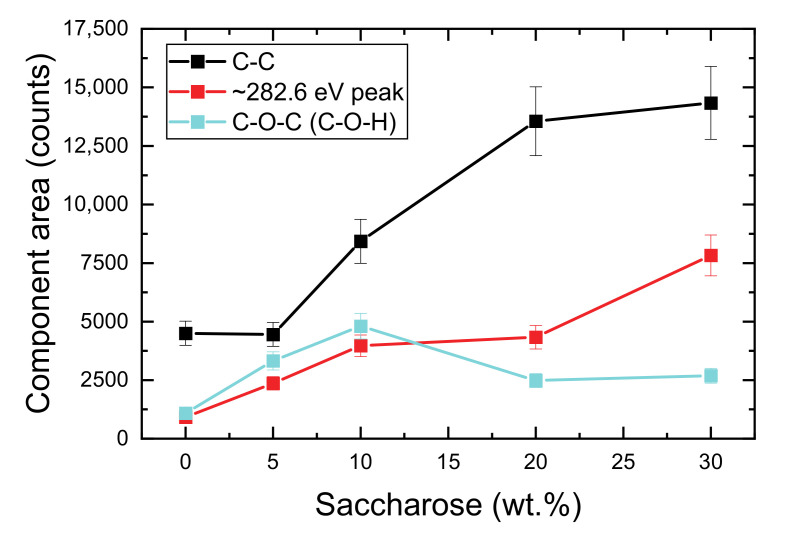
Areas of the C-O-C (C-O-H), C-C and ~282.6 eV components, obtained from the carbon C1s peak fit, presented as a function of the saccharose solution concentration (0—without modification, 5 wt%, 10 wt%, 20 wt%, 30 wt%).

**Figure 4 materials-13-05647-f004:**
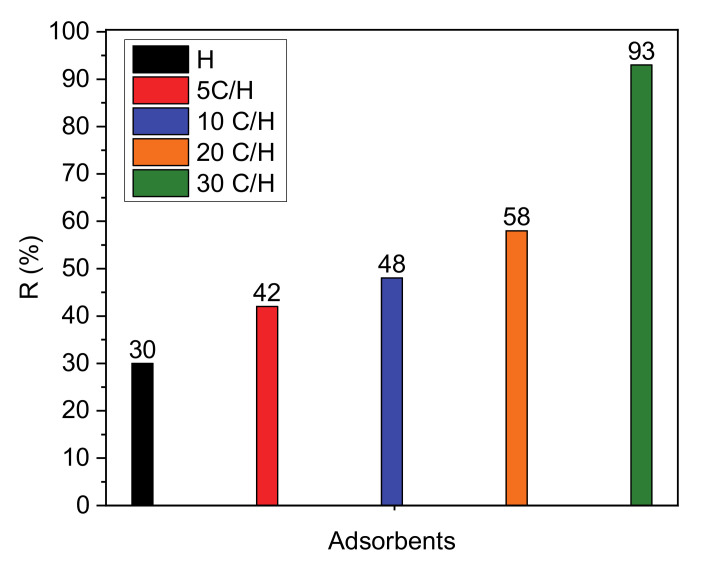
Efficiency of paracetamol removal for halloysite (H) and halloysite-carbon adsorbents (concentration of adsorbate solutions, 50 mg/dm^3^; mass of adsorbent, 0.5 g; temperature 25 °C).

**Figure 5 materials-13-05647-f005:**
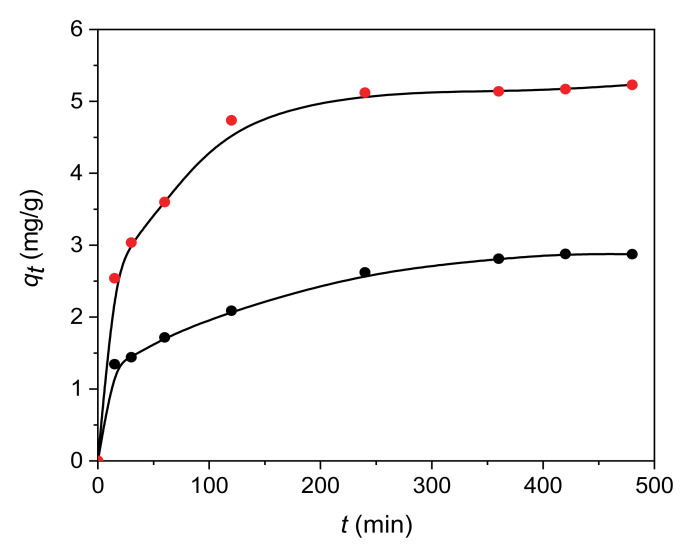
Kinetic adsorption curves of paracetamol on H (black circles) and 30 C/H (red circles) adsorbents.

**Figure 6 materials-13-05647-f006:**
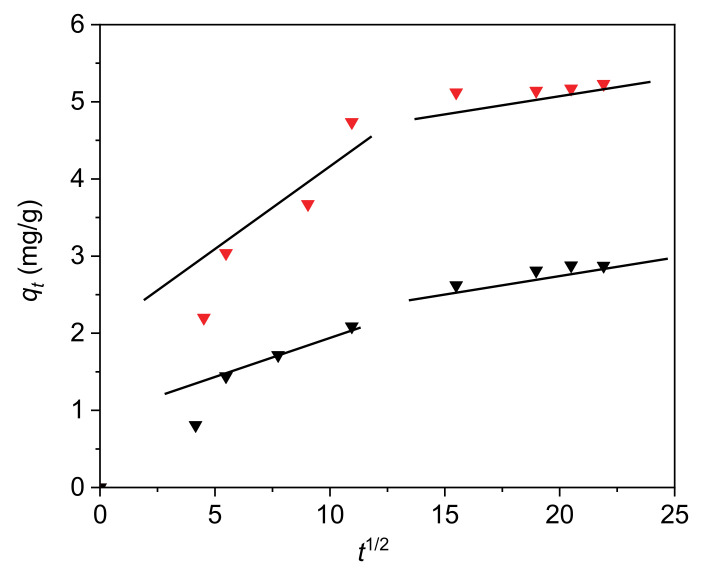
The intra-particle diffusion of paracetamol on H and 30/CH adsorbents (H—black triangles; 30C/H—red triangles).

**Figure 7 materials-13-05647-f007:**
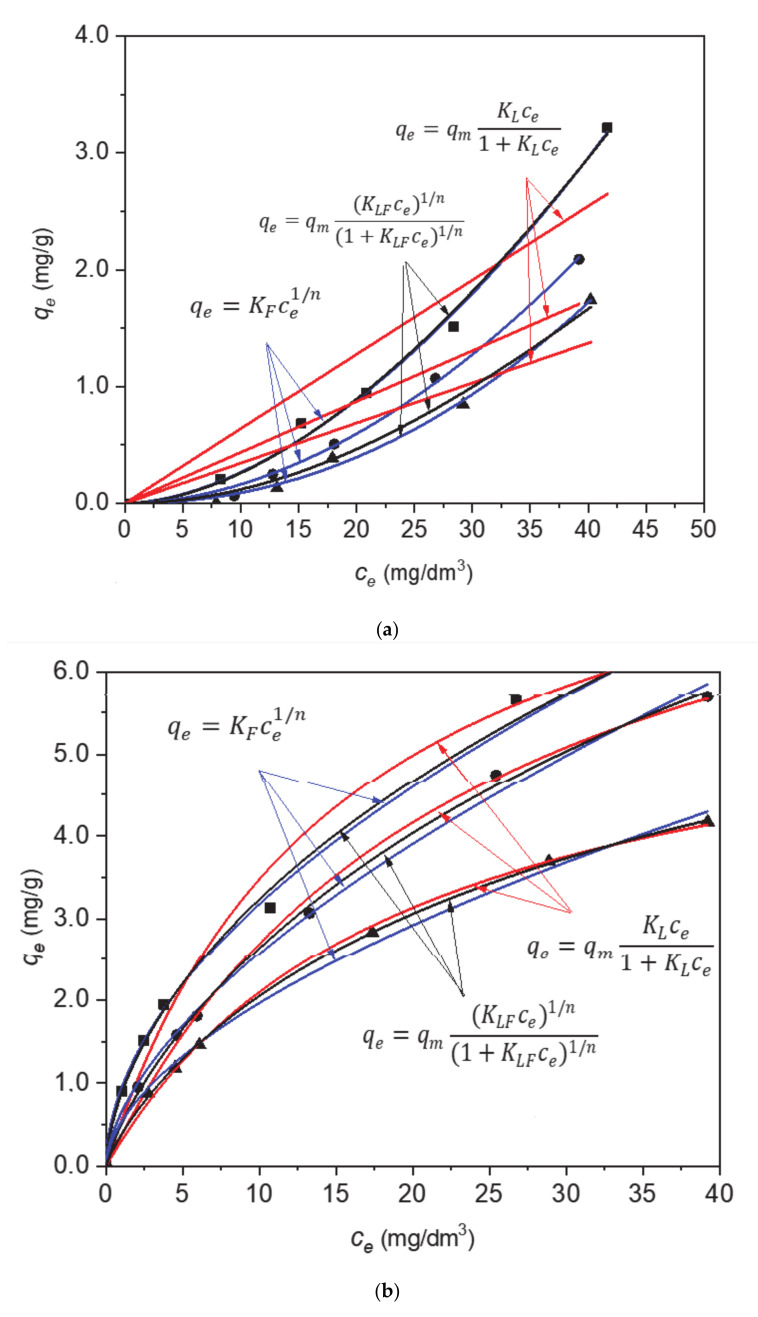
Calculation of adsorption equilibrium constant for paracetamol onto H (**a**) and 30C/H (**b**) adsorbents employing the experimental data. The lines represent the curve obtained by the application of the Freundlich and Langmuir (one-center and multi-center) equations with respect to the adsorption data as adjusted by the least-squares method. Temperature: ◼ 298 K; ⬤ 303 K; ▲ 313 K.

**Table 1 materials-13-05647-t001:** Atomic concentration (at%) of elements in the halloysite mineral for the H sample and the halloysite-carbon nanocomposites (5 C/H, 10 C/H, 20 C/H, and 30 C/H) determined using the XPS technique.

Element Sample	O (at%)	C (at%)	Si (at%)	Al (at%)	Fe (at%)
H	62.8	7.34	16.3	13.0	0.56
5C/H	59.5	9.92	16.4	13.8	0.47
10C/H	55.2	12.4	17.4	14.4	0.63
20C/H	54.7	15.9	15.3	13.0	1.15
30C/H	51.1	20.4	16.5	11.4	0.65

**Table 2 materials-13-05647-t002:** Result of the C1s peak decomposition into components corresponding to different carbon bonds, obtained for the purified halloysite sample and 5 C/H, 10 C/H, 20 C/H, and 30 C/H halloysite-carbon nanocomposites.

Component	FWHM (eV)	Position (eV)	Area	Area Uncertainty
**H sample**				
O-C=O	1.95	290.45	552.77	78.79
C=O	2.00	288.42	851.59	114.34
C-O-C (C-O-H)	2.00	286.83	1092.47	142.30
C-C	2.00	284.93	4499.66	517.05
~282.6 eV peak	2.20	282.64	915.42	121.80
**5C/H sample**				
O-C=O	2.00	289.51	407.51	60.94
C=O	2.01	287.97	1126.27	146.19
C-O-C (C-O-H)	2.01	286.17	3324.27	390.08
C-C	2.08	284.63	4453.26	512.06
~282.6 eV peak	2.12	282.63	2366.74	285.32
**10C/H sample**				
O-C=O	2.11	290.32	592.12	83.55
C=O	2.07	288.03	1503.63	189.14
C-O-C (C-O-H)	2.10	285.79	4809.26	550.27
C-C	2.07	284.53	8427.77	934.58
~282.6 eV peak	2.01	282.41	3968.63	459.86
**20C/H sample**				
O-C=O	2.11	290.61	795.15	107.71
C=O	2.10	288.39	1379.57	175.10
C-O-C (C-O-H)	2.07	286.39	2483.15	298.15
C-C	2.18	285.08	13,555.25	1471.95
~282.6 eV peak	2.05	282.72	4332.91	499.12
**30C/H sample**				
O-C=O	2.10	290.93	607.53	85.40
C=O	2.05	288.67	1429.30	180.74
C-O-C (C-O-H)	2.12	286.40	2690.53	320.92
C-C	2.17	284.82	14,335.89	1553.32
~282.6 eV peak	2.17	282.81	7829.35	871.42

**Table 3 materials-13-05647-t003:** The physicochemical properties of probe compounds.

Solvent Name	Cross-Sectional Area (m^2^)	Dispersive Energy of Probe Molecule (mJ/m^−2^)	*AN** (kcal/mol)	*DN* (kcal/mol)
Hexane	5.15 × 10^−19^	0.0184	-	-
Heptane	5.73 × 10^−19^	0.0203	-	-
Octane	6.3 × 10^−19^	0.0213	-	-
Nonane	6.9 × 10^−19^	0.0227	-	-
Acetone	3.4 × 10^−19^	0.0165	2.5	17
Acetonitrile	2.14 × 10^−19^	0.0275	4.7	14.1
Dichloromethane	2.45 × 10^−19^	0.0245	3.9	0
Ethyl Acetate	3.3 × 10^−19^	0.0196	1.5	17.1
Methanol	2.41 × 10^−19^	0.0181	12	20

**Table 4 materials-13-05647-t004:** Surface properties of carbon-halloysite samples.

Adsorbent	Specific Component of Free Energy Adsorption $-∆G_{a}^{sp}$ (kcal/mol)					*K_a_*	*K_b_*	*K_b_/K_a_*
	Acetone	Acetonitrile	Dichloro-Methane	Ethyl Acetate	Methanol			
H	3.9	2.7	1.1	2.9	4.8	0.16	0.22	1.38
5C/H	4.2	3.5	0.6	3.9	7.5	0.21	0.18	0.86
10C/H	2.6	1.9	0.8	3.2	6.5	0.16	0.11	0.68
20C/H	1.9	1.0	0.8	2.5	5.7	0.13	0.09	0.69
30C/H	1.7	0.9	0.7	2.2	4.3	0.11	0.06	0.54

**Table 5 materials-13-05647-t005:** Kinetic parameters of paracetamol adsorption on H and 30C/H adsorbents.

Adsorbent	Pseudo-First-Order Kinetic Model		Pseudo-Second-Order Kinetic Model	
	k_1_ (min^−1^)	R^2^	k_2_ (g mg^−1^ min^−1^)	R^2^
H	0.0136	0.8792	0.0552	0.9879
30C/H	0.0121	0.8665	0.0721	0.9992

**Table 6 materials-13-05647-t006:** Intra-particle diffusion model parameters.

Adsorbent	k_d1_ (mg g^−1^ min^−1/2^)	c_1_ (mg g^−1^)	R_1_^2^	k_d2_ (mg g^−1^ min^−1/2^)	c_2_ (mg g^−1^)	R_2_^2^
H	0.3263	0.91	0.8762	0.0341	1.77	0.9532
30C/H	0.5012	2.01	0.8456	0.0632	4.07	0.9651

**Table 7 materials-13-05647-t007:** Freundlich and Langmuir (one-center and multi-center) equation parameters, and correlation coefficients R^2^ for the adsorption of paracetamol on H and 30C/H adsorbents.

Isotherm	Parameter	Adsorbent					
		H			30C/H		
		Temperature (K)					
		298	303	313	298	303	313
Freundlich	*K_F_* (mg∙g^−1^) (dm^−3^∙mg^−1^)^1/n^	0.0048	0.0021	0.0006	0.8061	0.3730	0.2773
	*n*	1.72	1.87	2.12	0.57	0.66	0.69
	R^2^	0.8233	0.9245	0.9414	0.8945	0.8956	0.8922
Langmuir one-center	*K_L_* (dm^3^∙mg^−1^)	0.0673	0.0488	0.0352	0.0889	0.0684	0.0543
	*q_m_* (mg∙g^−1^)	7.8	6.0	5.7	8.7	6.8	6.1
	R^2^	0.9513	0.9321	0.9461	0.8734	0.8922	0.9432
Langmuir multi-center	*K_ML_* (dm^3^∙mg^−1^)^1/n^	0.0021	0.0018	0.0007	0.0487	0.0357	0.0291
	*q_m_* (mg∙g^−1^)	12.1	9.8	4.2	30.7	14.7	8.1
	*n*	0.50	0.56	0.55	1.6	1.28	1.20
	R^2^	0.9907	0.9984	0.9991	0.9962	0.9985	0.9994

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
