# Peer review of "Surface Properties of Halloysite-Carbon Nanocomposites and Their Application for Adsorption of Paracetamol"

_materials, 2020, doi:10.3390/ma13245647_

Round 1

Reviewer 1 Report

The authors have answered all the points considered

Author Response

Thank you for accepting the corrections in our manuscript.

We tried to improve the english language in our manuscript.

Reviewer 2 Report

The manuscript was revised by the authors once more. Some issues still need clarification and improvement, as follows.

Major issue:

  • Using the values of the specific component of the adsorption energy of the polar probes shown in Table 4, one cannot obtain (by equation 4) the values of Ka and Kb that are mentioned for 10C/H, 20C/H and 30C/H, although for H and 5C/H they seem correct.
  • In lines 229 and 236, the specific adsorption energy is not well defined. It is not RT ln(V’N,n) with RT ln(V’N,n) defined by equation 3.

Minor issues:

  • Abstract: a parenthesis is lacking in line 18 after “narrow scans”.
  • Units for DN and AN should be specified in Table 3. Kcal/mol?
  • The values presented for the molecular areas of the polar probes in Table 3 are much lower than those proposed by Riddle and Fowkes (1990), J Am Chem Soc 112:3259–3264 and more commonly used. Could the authors comment on this?
  • Lines 249-252. This text is repeated below in lines 267-270 and it should be removed from here.

Author Response

Answers to Reviewer

Comments and Suggestions for Authors

The manuscript was revised by the authors once more. Some issues still need clarification and improvement, as follows.

Major issue:

Question 1. Using the values of the specific component of the adsorption energy of the polar probes shown in Table 4, one cannot obtain (by equation 4) the values of Ka and Kb that are mentioned for 10C/H, 20C/H and 30C/H, although for H and 5C/H they seem correct.

Answer 1. The data in Table 4 for 10C/H, 20C/H and 30C/H were changed due to the conversion of the values of the specific component of the adsorption energy  ​​to the values expressed in kcal/mol.

Question 2. In lines 229 and 236, the specific adsorption energy is not well defined. It is not RT ln(V’N,n) with RT ln(V’N,n) defined by equation 3.

 Answer 2. We agree with the reviewer. This fragment of the manuscript has been changed.

Minor issues:

Question 1. Abstract: a parenthesis is lacking in line 18 after “narrow scans”.

Answer 1. This error was corrected.

Question 2. Units for DN and AN should be specified in Table 3. Kcal/mol?

Answer 2. The units for DN and AN were  specified in kcal/mol.

Question 3. The values presented for the molecular areas of the polar probes in Table 3 are much lower than those proposed by Riddle and Fowkes (1990), J Am Chem Soc 112:3259–3264 and more commonly used. Could the authors comment on this?

Answer 3. Cirrus Plus Software contains the physicochemical data of probe compounds. The values of AN* for polar probe compounds are the same as in Table III in the paper by Riddle and Fowkes (1990), J Am Chem Soc 112:3259–3264. These data are assigned to the computation in this program. Therefore, we used them for calculations. We explained in the manuscript how the values of AN* were calculated.

Question 4. Lines 249-252. This text is repeated below in lines 267-270 and it should be removed from here.

Answer 4. This error was corrected.

Round 2

Reviewer 2 Report

The manuscript was revised by the authors. I think that the manuscript can be accepted for publication after the following minor revisions:

rewrite lines 249-253, explaining better the calculation of AN*;

rewrite last sentence in lines 261-262;

check the values of Ka and Kb determined for sample 30 C/H.

Author Response

Answer to Reviewer 2

Question 1. Rewrite lines 249-253, explaining better the calculation of AN*.

Answer 1. A sentence has been added to explain better the calculation of AN*.

Question 2. Rewrite last sentence in lines 261-262.

Answer 2. This sentence was rewritten.

Question 3. Check the values of Ka and Kb determined for sample 30 C/H.

Answer 3. The values were corrected.

This manuscript is a resubmission of an earlier submission. The following is a list of the peer review reports and author responses from that submission.

Round 1

Reviewer 1 Report

This work demonstrates how the presence of carbon increases significantly the adsorption of halloysite-carbon composites for paracetamol. The authors have done a great job and the experiments are systematic and well designed. I just have some questions related to the XPS data.

The authors describe in the Materials and Methods section the use of a flood gun to compensate the charge one would expect for the samples studied. When explaining the XPS data obtained the authors describe that they did not do any correction of the binding energy scale for any of the data acquired using the adventitious carbon or any other contribution in the spectra They should be also clearly stated in the Materials and Methods section. 

They claim that they did not do any correction/calibration using for example the adventitious carbon contribution due to the difficulty in determining the location of this component before making the final fit. Although there is a lot of controversy in relation to use this adventitious carbon component,  i.e. the exact carbon species on a surface and thus the corresponding C 1s binding energies will always depend on the type and nature of the surface and the type and nature of the gases or vapours it has been exposed to, since the samples are prepared in similar conditions and were supposedly exposed under the same environment, a calibration to the “adventitious carbon peak” should  work relatively well to within some hundred meV. As an alternative, why the authors do not normalize the spectra after the fitting, correcting everything with this contribution as reference?

The different position of the components shown in Fig. 2 does not compromise the results and conclusions of this work, however, if the electron flood gun is used with all the samples under the same conditions and there are still minor charging effects, one would expect to see a shift towards lower energies when increasing the amount of C incorporated into the samples, and this is not the observed trend. Could the authors explain this fact?

When referring to the Fe impurities observed in the survey spectra, the authors should try to offer potential sources that explain this contamination observed.

In Fig.4, could the authors show the data for the "blank sample" H too?

English language and style are correct, however, minor spell checks are required. For example in lines 173-175, the authors write: 

"The results of the analysis are presented in the Fig. 2 for the H sample, whereas 5C/H, 10C/H, 20C/H, and 30C/H halloysite-carbon nanocomposites"

Reviewer 2 Report

This Manuscript reports on characterization of composites based on a nanoscale clay mineral halloysite and carbon produced via impregnation of the purified clay with saccharose and heating in nitrogen, and on adsorption of paracetamol. In comparison with previous publications of these authors, this contribution contains new data on surface properties the material and its potential application as adsorbent.

However, the MS suffers from a number of serious faults.

First of all, the interpretation of X-ray photoelectron spectra, and possibly the XPS experiment as a whole, are incorrect. Particularly, the authors discuss “C-C” species, making no difference between aliphatic carbon (sp3 state) with the binding energy of 285.0 eV and graphite (sp2) at about 284.5 eV that is expected to be the main reaction product of saccharide decomposition. Some other “C-C” species are also possible.

The formation of Si-C bonds looks unlikely from the chemistry point, with the line at about 282.6 eV being probably an artifact that is due to inhomogeneous electrostatic charging. This may be clarified by comparing the narrow scans of C with Si and other elements (Al, O, ..) but those are absent in the MS. Please also note that the component at 286.5 eV can be due to COH group rather than, or in addition to, C-O-C and the intensity above 290 eV is likely due to a satellite from graphite. Though, all the spectra fitting is inappropriate, if the charging took place.

In my opinion, the results of the experiments on IGC and adsorption of paracetamol should be reconsidered, and the text amended, keeping in mind that the main adsorbent in the composites, especially 30C/H, is activated carbon (graphite) rather than halloysite. This follows, in particular, from XPS analysis (Table 1). TEM images of the composites would be very useful as well, and it would be interesting to compare adsorption at the composites and a typical activated carbon sample.

Some other remarks:

Introduction

Please explain why adsorption of paracetamol was chosen to study and give its chemical name and formula.

The same wording. e.g. “from the “Dunino” strip mine (Legnica, Poland)”, “surface sensitive X-ray photoelectron spectroscopy (XPS)”, is repeated several times in neighboring sentences; please edit.

I am not sure that “the influence of halloysite surface changes on paracetamol adsorption” (line 74) is quite correct as a different, composite material was actually produced and used; please consider editing the wording.

Materials and methods

Please clarify the statement “carbonization at 800oC for 8 h, with a heating rate of 5oC/min under N2 atmosphere” (lines 86-87). Heating from room temperature to 800 oC takes about 3 h at this rate...     

The sentence in line 99-101 is wrong since the Ag 3d3/2 line width characterizes a resolution of the instrument but not the energy accuracy that depends also on a specific sample.  

The CasaXPS software should be mentioned in this section rather than in Results (line 154); moreover, details of the spectra processing are usually reported (peak shape, background and so on).

Consequently, I recommend a major revision of the MS, including Abstract, Conclusions and probably Title before its resubmission.

Author Response

Re: Effect of Surface Properties of Halloysite-Carbon Nanocomposites on the Adsorption Behavior of Paracetamol

by B. Szczepanik, D. BanaÅ›, A. Kubala-KukuÅ› et al. submitted to Materials

Dear Editor,

We would like to thank the referee for the work they have done to improve the quality of our article. The manuscript has been revised according to the reviewers’ remarks. Below is the list of corrections, modifications and replies to the comments.

Sincerely yours

Beata Szczepanik

Answers and list of changes:

Reviewer 2

  1. First of all, the interpretation of X-ray photoelectron spectra, and possibly the XPS experiment as a whole, are incorrect. Particularly, the authors discuss “C-C” species, making no difference between aliphatic carbon (sp3 state) with the binding energy of 285.0 eV and graphite (sp2) at about 284.5 eV that is expected to be the main reaction product of saccharide decomposition. Some other “C-C” species are also possible.

Answer: Of course. reviewer is right that our interpretation of the XPS photoelectron spectra is not very detailed, but in our opinion is good enough to support the conclusion of the article. Generally, in complex silicate structures a multitude of different peaks can be expected [*]. However for this article it goes too far to analyze each individual peak.

*Handbook of Mineral Spectroscopy Volume 1: X-ray Photoelectron Spectra, J. Theo Kloprogge, Barry J. Wood, March 2020

  1. The formation of Si-C bonds looks unlikely from the chemistry point, with the line at about 282.6 eV being probably an artifact that is due to inhomogeneous electrostatic charging. This may be clarified by comparing the narrow scans of C with Si and other elements (Al, O, ..) but those are absent in the MS.

Answer: The reviewer is right that sample charging is very important issue for measurement of halloysite samples because the sample has low conductivity. As a consequence of inappropriate charge compensation non-physical peaks related to the modification of the measured electron energy due to the sample charging can appear in the XPS photoelectron spectra. We have already investigated the influence of incorrect charge compensation on the XPS spectra in [22], and in our opinion the line about 282.6 eV which we interpreted as Si-C bond is not a result of inhomogeneous electrostatic charging. To confirm our conclusions we have added a new subfigure with narrow scans of Al and Si lines.

3) Please also note that the component at 286.5 eV can be due to COH group rather than, or in addition to, C-O-C and the intensity above 290 eV is likely due to a satellite from graphite.

Answer: Thank you for this note. We added information about this alternative possibility of the peak identification in the form C-O-C (or/and C-O-H) and corrected the table and figures accordingly. We also added sentence: “The intensity above 290 eV is likely due to a satellite from graphite.”

  1. In my opinion, the results of the experiments on IGC and adsorption of paracetamol should be reconsidered, and the text amended, keeping in mind that the main adsorbent in the composites, especially 30C/H, is activated carbon (graphite) rather than halloysite. This follows, in particular, from XPS analysis (Table 1). ?

Answer: we agree with the reviewer's opinion that main adsorbent in the studied composites is carbon rather than halloysite. IGC analysis of the composites and halloysite was carried out to show that the presence of carbon on the surface of halloysite changes the nature of this surface, significantly facilitating the adsorption of paracetamol on the surface of the composites.

  1. TEM images of the composites would be very useful as well, and it would be interesting to compare adsorption at the composites and a typical activated carbon sample.

Answer: It is difficult to compare adsorption on carbon-halloysite nanocomposites with adsorption on activated carbons. Activated carbons, depending on the carbon precursor and the method of preparation, may have very different specific surfaces, the degree of microporosity and adsorption properties in relation to such adsorbates as paracetamol. For example: Nche i in. reported the maximum capacities of activated carbons obtained from rice husk (SBET = 178 and 104 m2/g) about 15 -21 mg/g [1], while Nguyen et al. obtained the Langmuir maximum adsorption capacity of commercial activated carbon oxidized with HNO3 (SBET = 1284 m2/g) equal 221 mg/g [2].

[1] N.-A. George Nche, A. Bopda, D. R. Tchuifon Tchuifon, C. Sadeu Ngakou, I.-H. Tiotsop Kuete, A. S. Gabche, Removal of Paracetamol from Aqueous Solution by Adsorption onto Activated Carbon Prepared from Rice Husk, Journal of Chemical and Pharmaceutical Research, 2017, 9(3):56-68

[2] Nguyen D. T., Tran H. N., Juang R.-S., Duy Dat N., Tomul F., Ivanets A., Woo S. H., Hosseini-Bandegharaei A., Nguyen V. P., Chao H.-P. Adsorption process and mechanism of acetaminophen onto commercial activated carbon. J. Environ. Chem. Eng. 2020, in press, https://doi.org/10.1016/j.jece.2020.104408

  1. Introduction

Please explain why adsorption of paracetamol was chosen to study and give its chemical name and formula.

Answer: the following sentences were added in the Introduction:

„The presence of these pharmaceuticals in the environment is a growing problem because of their persistence and potential risk for the terrestrial and aquatic ecosystems.”

and „ We have selected paracetamol as a model compounds because it is a widely used over-the-counter analgesic and antipyretic drug, which have been detected in wastewater, surface waters, and drinking water throughout the world [21].

We added new position in References:

  1. Wu S., Zhang L., Chen J. Paracetamol in the environment and its degradation by microorganisms. Appl. Microbiol. Biotechnol. 2012, 96, 875–884.

The chemical name and formula of paracetamol were added in chapter 2.1. Materials and Reagents

  1. The same wording. e.g. “from the “Dunino” strip mine (Legnica, Poland)”, “surface sensitive X-ray photoelectron spectroscopy (XPS)”, is repeated several times in neighboring sentences; please edit.

Answer: It is corrected

  1. I am not sure that “the influence of halloysite surface changes on paracetamol adsorption” (line 74) is quite correct as a different, composite material was actually produced and used; please consider editing the wording.

Answer: this sentence was changed: „The studies were performed in order to investigate the influence of the surface properties of obtained nanocomposites on paracetamol adsorption from an aqueous solution in comparison to the unmodified halloysite”

  1. Materials and methods

Please clarify the statement “carbonization at 800oC for 8 h, with a heating rate of 5oC/min under N2 atmosphere” (lines 86-87). Heating from room temperature to 800 oC takes about 3 h at this rate...  

Answer: this fragment was changed “Halloysite-carbon composites were prepared through saccharose solution impregnation of halloysite and carbonization at constant temperature 800oC for 8 h (this temperature was obtained with a heating rate of 5oC/min) under N2 atmosphere”.

  1. The sentence in line 99-101 is wrong since the Ag 3d3/2 line width characterizes a resolution of the instrument but not the energy accuracy that depends also on a specific sample.  

Answer: The sentence was changed to the following:

„Full width in the half of maximum (FWHM) of the Ag3d 3/2 line measured by the system was about 1.0 eV. Energy calibration uncertainty measured with conducting Ag sample was not grater then 0.2 eV.”

  1. The CasaXPS software should be mentioned in this section rather than in Results (line 154); moreover, details of the spectra processing are usually reported (peak shape, background and so on).

Answer: The CasaXPS introduction was moved to this section and described as follows:

“Qualitative and quantitative analyses of the spectra were performed using the CasaXPS software (ver. 2.3) delivered with the SPECS XPS system. The photoelectron spectra were fitted using Shirley background and Lorentzian asymmetric lineshape with tail damping (LF).

Reviewer 3 Report

The manuscript “Effect of Surface Properties of Halloysite-Carbon Nanocomposites on the Adsorption Behavior of Paracetamol” analyses the surface properties of Halloysite-Carbon Nanocomposites and non-modified halloysite by XPS and IGC, and their adsorption behaviour of paracetamol. After a careful reading I have found several flaws in the manuscript, particularly the IGC part whose results are wrongly presented and discussed (it seems that the authors are not familiar with this technique). Thus, I cannot be in favour of the acceptance of the present manuscript. More details are provided by notes and suggestions marked in the pdf that goes in attached file.

Author Response

Re: Effect of Surface Properties of Halloysite-Carbon Nanocomposites on the Adsorption Behavior of Paracetamol

by B. Szczepanik, D. BanaÅ›, A. Kubala-KukuÅ› et al. submitted to Materials

Dear Editor,

We would like to thank the referee for the work they have done to improve the quality of our article. The manuscript has been revised according to the reviewers’ remarks. Below is the list of corrections, modifications and replies to the comments.

Sincerely yours

Beata Szczepanik

Answers and list of changes:

Reviewer 3

  1. This is not correct. This parameter is obtained by the vertical distances of the polar probes to the fitted line of the alkanes.

Answer: The error was corrected. The text has been completed: the vertical distance from the alkane line to the polar probe.

  1. A signal (-) is lacking in this equation, in account for the negative values of specific interactions.

Answer: The error was corrected.

  1. Where are the data for the dispersive component of surface energy of the different adsorbents? This section should be entirely revised.

Answer: The values of dispersive component of surface energy have been inserted into the Table 3.

  1. These are not the polar probes that were mentioned in the section 2.3 (line 110). This should be revised.

Answer: These errors were corrected.

  1. 5. Negative values of the Kb constants??? Values should be round to decimal point.

Answer:

In the Table 3 KB constants have positive values, the mistake has been removed.

  1. dispersive???? or specific?

Answer: In the table 3 the values of dispersive component of surface energy and dispersive component of free energy adsorption were presented. These errors were corrected.

  1. All this IGC section is wrong. Where is the dispersive component of the surface energy? It is not correct to obtain Ka and Kb values based on only two polar probes. Negative values of kb have no meaning. And if negative values are obtained, then analysing the ratio Ka/Kb makes no sense.

Use of these constants is very risky to assess the acid-base properties of the adsorbent and it is better compare specific affinities between acidic and basic probes. More polar probes should have been assessed (at least one acidic, one basic and one amphoteric).

For more details, authors may consult for instance:

Surface properties of carbonated and non-carbonated hydroxyapatites obtained after bone calcination at different temperatures, Colloids and Surfaces A, 2015, 478, 62-70.

Answer:

All of the responses to the reviewer comments have been corrected in the manuscript.

  1. 8. The part of IGC was not well addressed, as already mentioned before, and, accordingly, these conclusions are not correct.

Answer: The conclusions were extended and corrected as suggested by the reviewer.

  1. 9. 8. Three straight lines (red) for paracetamol adsorption on halloysite with langmuir (one center) model??? One curve for langmuir multicenter model seems to be missing.

Answer: Fit of adsorption equations (equation 5 in the manuscript) to the experimental data was completed with the Levenberg–Marquardt least-squares method using the Origin Microcal. The linear fit of equation 16 to the experimental data was obtained from Origin Microcal. It is probably caused the low concentrations of adsorbates.        

  1. 10. Language corrections and other minor fixes:

Line 96: in -> with

Line 101: grater then - > greater than

Figure 1: unit eV added

Line 151: In the figure 1, lines corresponding to Si 2p, 2s and Al 2p, 2s should be more clearly identified.

Line 151: H halloysite - > halloysite

Line 164: halloysite mineral for the H sample -> purified halloysite mineral sample

Line 174: whereas - > as well as for the

Line 185: detailed carbon spectra - > the narrow carbon scans

Line 185: adventitious/carbon -> adventitious carbon

Line 190: A result - > Results

Line 191: H purified -> purified

Line 197: in the - > as a

Figure 3: X-axis -> solutions deleted

Figure 3: O-C-O - > C-O-C

Answer: Corrected

Round 2

Reviewer 2 Report

The authors corrected the main body of reviewers’ remarks in the revised manuscript; however, the key problem with interpretation of XPS data still persists. It is obvious that SiC is absent in the composite as this substance forms at essentially higher temperatures than 800 oC used here, typically above 1300 oC, and it has not been detected by XRD and other techniques in a previous study of these researchers (ref. 22). So, I strongly suspect that the photoelectron C 1s line at about 282.5 eV is due to inhomogeneous electrostatic charging, particularly an excessive, negative charge arising at well-conducting portion of the composite flooded by slow electrons. Please note that the situation is more complicated than the charging of pure halloysite described in ref. 23 because the composite consists both of dielectric clay and conductive carbon. In addition, the Si 2p and other spectra in Fig.1 show no signs of silicon carbide (although those have not good enough resolution and so are not fully decisive).

Therefore, I could recommend to erase any mentioning of SiC in the paper. In principle, Table 2 and Figs 2 and Table 2 should be omitted too since the fits are strongly distorted by the inhomogeneous charging; though, the authors could try to explain the matter and to mitigate their claims about the forms of carbon. Maybe, the title of the paper is worth to be corrected as well. 

Author Response

Reviewer 2

Answers and list of changes:

  1. The authors corrected the main body of reviewers’ remarks in the revised manuscript; however, the key problem with interpretation of XPS data still persists. It is obvious that SiC is absent in the composite as this substance forms at essentially higher temperatures than 800 oC used here, typically above 1300 oC, and it has not been detected by XRD and other techniques in a previous study of these researchers (ref. 22). So, I strongly suspect that the photoelectron C 1s line at about 282.5 eV is due to inhomogeneous electrostatic charging, particularly an excessive, negative charge arising at well-conducting portion of the composite flooded by slow electrons. Please note that the situation is more complicated than the charging of pure halloysite described in ref. 23 because the composite consists both of dielectric clay and conductive carbon. In addition, the Si 2p and other spectra in Fig.1 show no signs of silicon carbide (although those have not good enough resolution and so are not fully decisive).

Therefore, I could recommend to erase any mentioning of SiC in the paper. In principle, Table 2 and Figs 2 and Table 2 should be omitted too since the fits are strongly distorted by the inhomogeneous charging; though, the authors could try to explain the matter and to mitigate their claims about the forms of carbon. Maybe, the title of the paper is worth to be corrected as well.

Answer: We have corrected the paper according to reviewer comments by mitigating our claims about the form of carbon correcting the following sentences:

“In the carbon peak spectra, the following components were identified: O-C=O, C=O, C-O-C (C-O-H), C-C, and Si-C. and ~282.6 eV component. The ~282.6 eV component corresponding to the C–Si species was identified in C1s spectrum registered for halloysite nanotubes after silane grafting [27]. However this finding requires further investigation of possible impact on the results of inhomogeneous electrostatic charging of the composite, consisting of both dielectric clay and conductive carbon.”

and “Also the positions of the other essential peaks identified agree well with the theoretical values C=O (288 eV), C-O-C (C-O-H) (286,5 eV) [28].”

  1. Kang H., Liu X., Zhang S., Li J., Functionalization of halloysite nanotubes (HNTs) via mussel-inspired surface modification and silane grafting for HNTs/soy protein isolate nanocomposite film preparation, RSC Adv., 2017, 7, 24140–24148.
  2. Ratner B. D., Castner D. G. (2009) Electron spectroscopy for chemical analysis. In: J. C. Vickerman (eds) Surface Analysis: The Principal Techniques, Wiley, Chichester, p. 56

We have also corrected the table and the figures accordingly.

  1. Maybe, the title of the paper is worth to be corrected as well.

The title was changed: “Surface Properties of Halloysite-Carbon Nanocomposites and their Application for Adsorption of Paracetamol”

Reviewer 3 Report

The manuscript “Effect of Surface Properties of Halloysite-Carbon Nanocomposites on the Adsorption Behavior of Paracetamol” was revised by the authors. However, some of the flaws already addressed in previous report by the referee, namely regarding the results of inverse gas chromatography and their discussion, are still present. Accordingly, I cannot recommend the acceptance of the present manuscript for publication. In detail:

  • The IGC section is full of mistakes and inconsistencies. The authors do not know how to distinguish dispersive and specific components of the surface energy: everything in Table 3 is shown as “dispersive”, which is unlikely. Furthermore, it is not correct to obtain Ka and Kb values based on only the assessment of two polar probes. For that, more polar probes should have been assessed (at least one acidic, one basic and one amphoteric). Notwithstanding, if only two polar probes are considered, it is then better to compare one to each other their specific affinities, where a typical Lewis acidic probe and a typical Lewis basic probe should be preferentially selected. Regarding this issue, the authors did not mention what were the values considered for DN and AN of acetonitrile and acetone, used for the calculation of Ka and Kb? Even the polar probes that have mentioned in the experimental section are different from those presented in the results of table 5. Apparently, the specific affinities of acetone (amphoteric) when comparing H and 5C/H are similar and the same for acetonitrile (acidic), but Ka and Kb values are completely different between H and 5C/H? This is very strange conclusion. Authors also did not comment the values obtained for dispersive component of the surface energy. Nothing was explained. Accordingly, all the statements mentioned at page 11 for IGC are doubtful. This also applies for conclusions and abstract sections in what concerns the statements on IGC.

Other issues

  • Title: Authors have not answered to the issue previously addressed in the latest reviewer report “This title does not reflect the manuscript content, because no correlations have been demonstrated between the surface properties assessed by IGC and XPS and the results of paracetamol adsorption.”
  • In the graphs of figure 1 that was revised, the scale of the survey XPS spectrum (top panel) is not the same as that of the spectrum shown in the bottom panel. In fact, no scale was provided for the top panel spectrum.
  • The results for Ci in figure 7 and table 5 still do not match. For instance, in the first linear part of the graph for halloysite, C1 (intercept) in the graphic (Figure 7) is around 0.5, but the value shown in the table is 0.91! The same applies for the other values of Ci.

Author Response

Reviewer 3

Answers and list of changes:

  1. The IGC section is full of mistakes and inconsistencies. The authors do not know how to distinguish dispersive and specific components of the surface energy: everything in Table 3 is shown as “dispersive”, which is unlikely.

Answer: These mistakes were corrected.

  1. Furthermore, it is not correct to obtain Ka and Kb values based on only the assessment of two polar probes. For that, more polar probes should have been assessed (at least one acidic, one basic and one amphoteric). Notwithstanding, if only two polar probes are considered, it is then better to compare one to each other their specific affinities, where a typical Lewis acidic probe and a typical Lewis basic probe should be preferentially selected. Regarding this issue, the authors did not mention what were the values considered for DN and AN of acetonitrile and acetone, used for the calculation of Ka and Kb? Even the polar probes that have mentioned in the experimental section are different from those presented in the results of table 5. Apparently, the specific affinities of acetone (amphoteric) when comparing H and 5C/H are similar and the same for acetonitrile (acidic), but Ka and Kb values are completely different between H and 5C/H? This is very strange conclusion.

Answer: As suggested by the reviewer, polar probes with acidic, amphoteric and basic properties were added in the corrected text. Figure 4 and Table 3 were changed as suggested by the Reviewer

  1. Authors also did not comment the values obtained for dispersive component of the surface energy. Nothing was explained. Accordingly, all the statements mentioned at page 11 for IGC are doubtful. This also applies for conclusions and abstract sections in what concerns the statements on IGC.

Answer: The comment concerning the values obtained for dispersive component of the surface energy was added in the corrected text.

  1. Title: Authors have not answered to the issue previously addressed in the latest reviewer report “This title does not reflect the manuscript content, because no correlations have been demonstrated between the surface properties assessed by IGC and XPS and the results of paracetamol adsorption.”

Answer: The title was changed: “Surface Properties of Halloysite-Carbon Nanocomposites and their Application for Adsorption of Paracetamol”

  1. In the graphs of figure 1 that was revised, the scale of the survey XPS spectrum (top panel) is not the same as that of the spectrum shown in the bottom panel. In fact, no scale was provided for the top panel spectrum.

Answer: The scale for the top panel is above the panel

  1. The results for Ci in figure 7 and table 5 still do not match. For instance, in the first linear part of the graph for halloysite, C1 (intercept) in the graphic (Figure 7) is around 0.5, but the value shown in the table is 0.91! The same applies for the other values of Ci.

Answer: The graph on Figure 7 was corrected

Round 3

Reviewer 2 Report

The requested corrections have been made; I believe that the paper can be published now

Reviewer 3 Report

The manuscript was revised by the authors. However, there is high inconsistencies regarding the IGC part when comparing the results presented in the Table 3 for specific interactions and the corresponding plots of figure 4. Accordingly, I cannot recommend the acceptance of the present manuscript for publication:

  • The IGC section is not consistent and not correct: the values of the specific component of free energy of adsorption of the polar probes on the surface of materials shown in Table 3 do not match with the plots of Figure 4. For instance, for acetonitrile in 5C/H, from the plot of Figure 4, a value of specific interaction around 3 kJ/mol comes from the graphic (vertical distance from the point to the reference line), whereas the value shown in the Table 3 is 14.9 kJ/mol! How is this possible? And this applies for all the values shown in table 3. Were the calculations well done?
  • The authors should present (in a new table) the values of the reference parameters considered for all the tested probes, including the molecular surface area and the dispersive component of surface tension of all the probe molecules. The values for DN and AN of all the tested polar probes (methanol, acetone, acetonitrile, ethyl acetate, and dichloromethane), used for the calculation of Ka and Kb parameters, should be presented as well. Otherwise, no one will be able to reproduce the calculation of these parameters, whose values presented in Table 3 raise doubts.
  • The results for C2 in Figure 7 (intercept) and Table 5 still do not match.